# Plocabulin Displays Strong Cytotoxic Activity in a Personalized Colon Cancer Patient-Derived 3D Organoid Assay

**DOI:** 10.3390/md17110648

**Published:** 2019-11-19

**Authors:** Alba Costales-Carrera, Asunción Fernández-Barral, Pilar Bustamante-Madrid, Laura Guerra, Ramón Cantero, Antonio Barbáchano, Alberto Muñoz

**Affiliations:** 1Department of Cancer Biology, Instituto de Investigaciones Biomédicas “Alberto Sols” (CSIC-UAM) 28029 Madrid, Spain and University Hospital La Paz Institute for Health Research (IdiPAZ), 28046 Madrid, Spain; albacostales@iib.uam.es (A.C.-C.); afbarral@iib.uam.es (A.F.-B.); pbustamante@iib.uam.es (P.B.-M.); abarbachano@iib.uam.es (A.B.); 2Biomedical Research Networking Centres-Oncology (CIBERONC), 28029 Madrid, Spain; 3General Surgery and Pathology Services, University Hospital La Paz, 28046 Madrid, Spain; laura.guerra@salud.madrid.org (L.G.); ramon.cantero@salud.madrid.org (R.C.)

**Keywords:** plocabulin, drug assay, patient-derived organoids, 3D culture, colorectal cancer

## Abstract

Plocabulin is a novel microtubule-disrupting antitumor agent of marine origin that is currently undergoing phase II clinical trials. Plocabulin has potent antiproliferative and antiangiogenic actions in carcinoma cell lines and has antitumor activity in xenografted mice. Here, we used three-dimensional (3D) tumor organoids derived from three colorectal cancer (CRC) patients to study the effect of plocabulin in a personalized assay system that ensures dose dependence and high reproducibility. The cytotoxicity of plocabulin was an order of magnitude higher than that of the active irinotecan derivative SN38 (7-ethyl-10-hydroxy-camptothecin) in tumor organoids at different passages. Moreover, plocabulin maintained its strong cytotoxic activity in wash-out experiments, in which a short pulse treatment of tumor organoids was as efficient as continuous treatment. Our data show that plocabulin has a very potent cytotoxic action in CRC patient-derived tumor organoids, supporting ongoing clinical trials with plocabulin and the use of organoid assays to provide personalized validation of antitumor drugs.

## 1. Introduction

Plocabulin (PM060184, C_31_H_45_N_3_O_7_) is a member of a new family of tubulin-binding agents that has completed phase I studies and is undergoing phase II clinical trials for the treatment of solid tumors [1,2]. Originally isolated from the Madagascan marine sponge *Lithoplocamia lithistoides*, plocabulin is now obtained by total synthesis [3]. Plocabulin binds tubulin with high affinity, reducing its polymerization and depolymerization to a similar extent and thereby inhibiting microtubule dynamicity (Figure 1) [4,5,6]. This action translates into a strong antiproliferative effect on cancer cells growing in 2D cultures, an antiangiogenic effect on endothelial cells in vitro, and antitumor activity in xenografted mice [5,7].

Organoids are emerging as a promising preclinical 3D model of human disease and an advantageous system for assaying antitumor drugs [10]. For decades, most compounds that showed cytotoxic activity in the classical test panel of cancer cell lines failed in the early phases of clinical studies [11]. This low success rate was attributed to dissimilarity between patient tumors and genetically unstable immortal cell lines after prolonged culture. In contrast, organoids are 3D structures generated in culture from primary stem cells isolated from human healthy or tumor tissue and can be used to test drugs soon after isolation. Patient-derived tumor organoids recapitulate the histologic, genetic, and molecular features of the tumors from which they are derived and show high genome stability [12]. They are therefore a suitable complement to tumor genome sequencing and xenograft studies for the prediction of patient responses to treatments [13,14,15,16,17,18]. Initial proof-of-concept studies by van de Wetering et al. [19] demonstrated the validity of patient-derived colorectal cancer (CRC) organoids for testing many anticancer drugs in robotized procedures, although reproducibility remained an unresolved issue. Clearly, the future utility of organoids in personalized medicine will depend on the concordance between organoid responses to drugs and patient responses to the same drugs in clinical practice. Organoids have the potential to reduce inefficient medication, deleterious side-effects, and health care costs. Organoid use was first supported by a study with organoids from 23 gastrointestinal cancer patients, showing 88% positive predictive value and, importantly, 100% negative predictive value [20]. More recently, research has extended to other types of cancer, demonstrating the feasibility of patient-derived organoids for treatment testing and as a promising tool for predicting patient responses [15,16,17,18].

To evaluate the activity of plocabulin against colon cancer in a physiologically relevant setting, we investigated its cytotoxic activity in 3D organoids generated from tumor tissue obtained from three CRC patients from our Living Biobank recently described in [21]. In this study, we directly generated a panel of colon organoids from fresh surgical biopsies obtained from CRC patients. These organoids were used to analyze the effects of vitamin D on normal and cancer stem cell gene expression, proliferation, and differentiation [21]. For comparison with plocabulin, the same tumor organoid cultures were treated in parallel with SN38, an in–vivo-generated derivative of the topoisomerase I inhibitor irinotecan that is widely used to treat CRC. To obtain valid reproducible data from the cytotoxicity assays, we first optimized the organoid assay for each patient by selecting the number and size of organoids to seed. Our results reveal strong dose-dependent cytotoxic activity of plocabulin, with an IC_50_ in the low nanomolar range, which compares favorably with SN38 activity. Moreover, pulse treatments of patient-derived organoids revealed potent plocabulin cytotoxicity in wash-out assays.

## 2. Results

### 2.1. 3D Organoid Culture and Optimization of Drug-Assays

Organoid cultures were generated from tumor-tissue biopsies obtained during surgery from therapy-naive CRC patients in our previously described Living biobank [21]. Organoids had the usual spherical morphology, with variably sized diameters and lumens (Figure 2A). As expected, tumor organoids grew in the absence of Wnt3A (Wingless-Type mouse mammary tumor virus integration site family 3A), R-Spondin, and nicotinamide. These factors are required for growth of organoids derived from healthy tissue but are dispensable for the growth of colon tumor organoids because these harbor mutations (*APC*, *CTNNB1* (β-catenin), or *AXIN*) that activate Wnt signaling independently of ligands. We first tested plocabulin activity in a pilot experiment to ensure that the highly compact nature of organoids did not block drug access to the tumor cells. In line with its reported mechanism of action, plocabulin caused dose-dependent disorganization of the cellular microtubule system and of organoid structure (Figure 2B). 

To ensure reproducibility of results in cytotoxicity assays, for each patient organoid culture we previously defined the optimal size and number of organoids to seed (Figure 3A,B). This optimization procedure defined precise parameters that differ between patients. Organoids from patients #3, #4, and #29 had the same optimum assay parameters (20–40 μm diameter organoids and 250 organoids/well seeding density) and were selected for the cytotoxicity assays. Whereas small organoids (Figure 3B, purple) generated insufficient signal and large organoids (red) showed a non-linear response due to incomplete lysis, medium-sized organoids (green) showed a linear relationship between luminosity and organoid number and also yielded the highest Z-score (Figure 3B and Appendix A). The Z-score criterion was adapted for drug assays [22,23], ensuring reliable detection of drug effects. 

### 2.2. Plocabulin Cytotoxic Activity in Patient-Derived Tumor Organoids

The clinicopathological characteristics of patients #3, #4, and #29 are shown in Figure 4A. Mutational analysis showed that the organoids harbored genetic alterations typical of sporadic CRC: *APC* (3/3), *TP53* (2/3), *KRAS* (1/3), and *NRAS* (1/3) (Figure 4B). 

To promote optimum organoid growth and allow medium replacement and washes, the organoids were embedded in 70% Matrigel instead of the low (2%) concentration used in robotized screening studies (cytotoxicity assay design in Figure 5A) [19]. Cytotoxicity assay quality was also checked by analyzing the Z-score. For example, patient #4 organoid cultures in three independent assays (passages 25, 27 and 35) were incubated with increasing doses of plocabulin for 4 days (Figure 5B, representative images in 5C). This treatment time was chosen as it is the longest period during which the organoids can maintain growth with no medium change. This enhances growth differences between the control and high-dose treatment conditions and thus yields better/higher Z-score values. In this and all other assays, the Z-score was above 0.5 (excellent) (Figure 5B and Appendix A). Organoids at different passages were used to test data reproducibility over time in culture, thus providing validation of organoid stability and results.

We studied the response of patients #3, #4, and #29 organoid cultures to a 4-day treatment with different plocabulin doses. Doses were run in triplicate (assay design in Figure 5A), and assays were repeated three times, using organoids at different passages (Appendix A). Dose-responses (dropping steeply between 0.1 nM and 2 nM) and IC_50_ values were very similar for all patient-derived organoid cultures (patient #3, 1.1 nM; patient #4, 0.9 nM; and patient #29, 0.7 nM) and showed high inter-experiment reproducibility in organoids at different passages (Figure 6A). Notably, parallel assays in all organoid cultures showed a weaker response to SN38 (7-Ethyl-10-hydroxy-camptothecin), with IC_50_ values (half maximal inhibitory concentration) an order of magnitude higher (patient #3, 44.0 nM; patient #4, 28.5 nM; and patient #29, 66.8 nM) (Figure 6A). A combined plot of all results provides a clearer visualization of the responses to the two agents (Figure 6B). These results show that plocabulin has 30 to 95 times more potent cytotoxic activity than SN38 in the studied organoid cultures.

We also investigated the activity of plocabulin in wash-out experiments, comparing the response of tumor organoids to several plocabulin doses during a 4-day continuous treatment or a 4-h pulse treatment (Figure 7A and Appendix A). IC_50_ values in the pulse treatments were only moderately higher (less than an order of magnitude) than in the continuous treatment assays (Figure 7B).

## 3. Discussion

Our results demonstrate that the novel anticancer agent of marine origin plocabulin has potent cytotoxic activity in human tumor organoids derived from CRC patients. In cultures of tumor organoids derived from three therapy-naive individuals, plocabulin was more cytotoxic than SN38, the active derivative of irinotecan, a drug that is widely used to treat CRC. 

Organoids are a new 3D model system for testing anticancer drugs that has many advantages over traditional immortal cancer cell lines and xenografted animals [25]. Organoids recapitulate features of patient tumors in vivo (such as the histology and mutational landscape) better than 2D cultures of cell lines growing on plastic; moreover, in terms of drug sensitivity, organoids provide personalized data of clinical utility. Compared with patient-derived xenografts in mice, organoid cultures are easier to establish, require less specialized facilities, save animal testing, have a lower cost to benefit ratio, and are more amenable to low-throughput drug screening [10,25]. However, a key limitation of drug activity assays using organoids is reproducibility. Limitations and potential problems include firstly, an incomplete cell disaggregation of organoids when they are split, which can affect cell counting, and secondly, the high organoid heterogeneity derived from the non-clonal origin of the cultures. We addressed this problem by optimizing assay conditions in a personalized/individual way to establish the most appropriate organoid size and seeding density to ensure maximum Z-score. Similarly, after pilot tests, we selected a series of seven doses and 4-day treatment as the optimal conditions to obtain a full range of drug activity (from no effect to maximum cytotoxicity) and IC_50_ values. Quality control of the assays confirmed the validity of the selected conditions, and cytotoxicity assays were highly reproducible in organoids at different passages. Although not applicable to high-throughput screenings, our protocol proved to be highly efficient for ensuring quality in the study of a low number of drugs in secondary screenings. Current technology allows to generate from the same patient organoids from healthy and tumor tissues, which can be useful to initially analyze toxicity of drugs at earlier stages of development than plocabulin.

Previous studies of colon carcinoma cell lines grown in 2D cultures had provided IC_50_ values for plocabulin in the low nanomolar range (LoVo, 0.14 nM; HT29, 0.04 nM; HCT116, 4.6 nM), similar to those obtained in our organoids [4]. These cell lines harbor mutations that similar to those found in our organoids, such as activating mutations of Wnt signaling (*APC*, *CTNNB1*/β-catenin or *AXIN2*) and TGF-β signaling (*SMAD2/4*) and others affecting crucial CRC genes (*KRAS* and/or *BRAF* and *TP53*). 

Interestingly, our data from wash-out experiments favor a hit-and-run effect of plocabulin that is compatible with its proposed mechanism of action: High affinity binding to tubulin leading to the disruption of the microtubule cytoskeleton. This type of wash-out assay mimics physiological drug-clearance in the organism, and amenability to this analysis is another advantage of our protocol versus standard procedures, which do not allow medium changes due to the use of very low Matrigel concentrations that impede solidification.

Plocabulin has a unique mechanism of microtubule disruption and the highest affinity among tubulin-binding agents [3,4]. It shows strong activity in cancer cell lines in vitro and in xenografted animals [5,7]. Remarkably, plocabulin showed clinical benefits (33% rate) in a phase I trial in patients with a variety of cancer types [1]. Plocabulin is currently undergoing phase II trials in advanced or metastatic CRC after standard treatment (NCT03427268). Our results in CRC patient-derived tumor organoids reinforce these data, increasing interest in this novel anticancer agent and encouraging further studies, particularly in CRC; these studies will need to be paralleled by analysis of plocabulin pharmacology and toxicity in these patients.

## 4. Materials and Methods 

### 4.1. Human Samples and Ethical Guidelines

Fresh human tissues were provided by IdiPAZ Biobank, part of the Spanish Biobank Network (www.redbiobancos.es), from individuals diagnosed with left colon cancer and subjected to surgery. All human patients gave informed consent. Biopsy histology was evaluated by the La Paz University Hospital Pathology Service. The study complied with the Declaration of Helsinki regarding human participants and was approved by the Ethics Committee of La Paz University Hospital (CEIC, HULP-PI-1425). Confidentiality of sample-related data is guaranteed under current legislation (Regulation 2016/679 of the European Parliament and of the Council of April 27, 2016 on Data Protection (GDPR), Biomedical Research Law 14/2007 and Royal Decree of Biobanks 1716/2011). All documents were approved by the hospital CEIC (Comité Ético de Investigación Clínica/Clinical Research Ethics Committee), and all team members are committed to maintaining confidentiality. 

### 4.2. Reagents

Plocabulin was obtained by total synthesis [3] at PharmaMar (Madrid, Spain), prepared as a 1 mg/mL stock solution in DMSO (dimethyl sulfoxide), and stored at –80 °C. SN38 (Tocris, # 2680) was prepared as a 10 mM solution in DMSO and stored at –20 °C. The vehicle in drug assays was DMSO.

### 4.3. 3D Tumor Organoid Cultures

Human tumor organoid cultures were generated as described [21]. Briefly, human biopsies were washed several times in PBS and incubated with a mixture of antibiotics for 30 min. The tissue was cut into small pieces and digested with collagenase for 30 min at 37 °C. Cells were disaggregated by passing the suspension through an 18G syringe needle. Single cells were collected by passing the solution through a 70 μm mesh filter into a 50 mL tube followed by centrifugation at 250× *g* for 5 min at 4 °C. For erythrocyte lysis, the cells were incubated in 157 mM NH_4_Cl for 5 min, washed in PBS, centrifuged and re-washed in washing buffer (Advanced DMEM/F12 basal medium (GIBCO, Grand Island, NY, USA) supplemented with 10 mM HEPES, 10 mM Glutamax (GIBCO, Paisley, Scotland, UK). Pelleted cells were embedded in Matrigel and seeded on pre-warmed 6-well culture plates. After Matrigel solidification, culture medium was added (Advanced DMEM/F12 basal medium supplemented with 10 mM HEPES, 10 mM Glutamax, 1 × N2 supplement, 1 × B27 supplement (GIBCO, Grand Island, NY, USA), 1 mM *N*-acetyl-l-cysteine (Sigma-Aldrich, Saint Louis, MO, USA), 1:500 Primocin, 0.1 μg/mL Noggin (Peprotech Rocky Hill, NJ, USA), 1 μg/mL Gastrin (Tocris, Bristol, UK), 50 ng/mL EGF (Peprotech, Rocky Hill, NJ, USA), 0.02 μM PGE_2_ (Sigma-Aldrich), 1 μM LY-2157299 (Axon-Medchem, Groningen, The Netherlands), and 10 μM SB-202190 (Sigma-Aldrich, Saint Louis, MO, USA).

### 4.4. Immunofluorescence

Tumor organoids embedded in Matrigel were washed twice in PBS and collected in a biopsy cassette Tissue-Tek Paraform (Sakura) for paraffin blocks. Organoids were fixed with 4% PFA and incubated with mouse anti-alpha-tubulin antibody (1:800) (Sigma, T5168) and secondary goat anti-mouse antibody (1:1000) (Life Technologies, A11029, Rockford, IL, USA). Images were taken using a DM2000 Leica microscope equipped with the LAS AF software (version 2.6.0.7266, Leica, Heidelberg, Germany).

### 4.5. Growth and Expansion of Organoid Cultures

Culture medium was changed every 2–3 days. For passaging, Matrigel-embedded organoids were collected by scraping and incubated on ice in Cell Recovery Solution (Corning) for 30 min. After centrifugation, organoids were washed in washing buffer and centrifuged again. Organoids were then incubated in washing buffer containing 1 mg/mL Dispase for 10 min at room temperature. Immediately afterwards, EDTA was added to a final concentration of 2 mM, and the mixture was incubated for 5 min. Disaggregated organoids were homogenized by passing through a 21G syringe needle, washed twice in washing buffer, embedded in Matrigel, and seeded in 6-well culture plates.

### 4.6. Mutational Analysis

Matrigel was removed using ice-cold Cell Recovery Solution as described above. DNA was extracted from pelleted organoids by overnight incubation at 56 °C in lysis buffer (50 mM TrisHCl pH 8.0, 100 mM EDTA pH 8, 100 mM NaCl, 1% SDS, and 20 mg/mL proteinase K (Merck, Darmstadt, Germany)). Saturated NaCl was added for five min, and DNA was precipitated with isopropanol and washed twice in 70% ethanol. Mutations in tumor organoid cultures from CRC patients #3 and #4 (passages 5 and 4) were analyzed by sequencing the amplified product of a multiplexed-PCR reaction (Amplicon sequencing) using a proofreading polymerase in an Illumina MiSeq instrument, as described (15). Mutations in tumor organoids from patient #29 (passage 25) were analyzed by sequencing the amplified product of a multiplexed-PCR using the Ion AmpliSeq™ Library Kit 2.0 and Ion AmpliSeq™ Cancer Hotspot Panel v2 (Life Technologies). For PCR, a total of 20–24 cycles was used for each sample. PCR templates were prepared and enriched using the Ion PGM Hi-Q OT2 Kit and the Ion OneTouch 2 System. Finally, the Ion PGM™ Sequencing 200 Kit v2 and Ion PGM™ System (Life Technologies) were used for DNA sequencing according to the manufacturer’s protocols. Alignment to the Hg19 human reference genome and variant calling were performed with Torrent Suite™ Software v.4.2.1 (Life Technologies). The mean coverage per sequenced sample was approximately 1000 reads per base. Variants with a Phred-score quality field value below 100 were considered low-quality variants. The effect of genomic variants on protein function was predicted using the PROVEAN Genome Variants tool (http://provean.jcvi.org/index.php) [26]. Variants with a possibly damaging or deleterious effect predicted by at least one PROVEAN predictor were considered of interest and visually checked with the Integrative Genomics Viewer (IGV) v.2.3.40, Broad Institute [27]. Variants with global minor allele frequencies above 0.05 were considered single nucleotide polymorphisms and were rejected (data from dbSNP, http://www.ncbi.nlm.nih.gov/SNP/). 

### 4.7. Organoid Size Variability

Pictures of organoid culture drops were taken 6 days after seeding, and organoids in each drop were counted and measured using MIPAR software (version 2.0, MIPAR, Worthington, OH, USA). The image-processing recipe included 28 specifically tailored steps for counting organoid number and area (provided on request). An example of the MIPAR counting recipe is shown in Figure 2A: small organoids (<30 μm) are highlighted yellow, medium-sized organoids (31–149 μm) cian, and large organoids (>150 μm) red. Phase-contrast images were captured with a DFC550 digital camera (Leica, Wetzlar, Germany) mounted on an inverted TS100 microscope (Nikon, Tokyo, Japan).

### 4.8. Optimization of Drug Activity Assays

Each patient-derived organoid culture has a specific growth pattern. To perform a reliable cell viability assay in organoids exposed to drugs, for each patient it was necessary to optimize two parameters: a) organoid size and b) number of organoids seeded per well. The objective is to ensure a proportional response between luminosity units and the number of seeded cells (Figure 3B). For this, Matrigel-embedded organoids were collected in washing buffer and filtered successively through 70 μm, 40 μm, and 20 μm strainers (Pluriselect) to obtain three sized organoid sets: 0–20 μm; 20–40 μm; and 40–70 μm (Figure 3A). The three sets of size-selected organoids were resuspended in 500 μL culture medium, and their density was estimated and adjusted by manual counting of 2 μL drops (Eight replicates at different dilutions). Combinations of different sizes and concentrations of organoids were seeded in triplicate in MW96 (2% Matrigel, total volume 100 µL) and incubated with vehicle (0) and with increasing concentrations of plocabulin (Figure 3B). Plocabulin was used as a killing agent to calculate quality-control Z-scores and ensure good quality control parameters for the subsequent cytotoxicity assays. Four days later, 100 μL of Cell Titer 3D reagent was added to each well, the plate was incubated in the dark on an orbital shaker at maximum speed for 5 min and left to rest in the dark for 30 min. Luminosity was measured three times in a Glomax 96 luminometer (Promega). The optimal number of organoids for seeding was chosen according to the proportional intensity of the luminous signal relative to the number of organoids seeded, the minimum error among replicates, and the best fitting Z-score. For patients #3, #4, and #29 from our biobank (15), these tests established 20–40 μm diameter and 250 organoids/well as the optimal seeding conditions for cytotoxicity assays.

### 4.9. *Z-score*


Z-score was used as the quality control parameter for the optimization of organoid assays and cytotoxicity assays [22,23]. It is calculated as follows:
$$Z-score=\;1-\frac{\left(3\;\times \;stDev\;of\;MaxDose\;+\;3\;\times \;stDev\;of\;Control\right)}{\left(Mean\;Control\;-Mean\;MaxDose\right)}$$

As is clear from the formula, the Z-score can never exceed 1. Values between 0.5 and 1.0 indicate an excellent assay; values from 0.4 to 0.5 indicate a decent assay; and values below 0.4 indicate a marginal assay. A value below 0 indicates too much overlap between the control and the maximum dose values for the assay to be useful.

### 4.10. Cytotoxicity Assays

Full-grown organoids were collected by scraping and placed in 15 mL tubes. Matrigel was removed by incubation in Cell Recovery Solution for 30 min on ice with continuous shaking. After a wash in cold PBS and centrifugation at 260× *g* for 5 min at 4 °C, organoids were resuspended in washing buffer and were size-selected using 20 and 40 μm cell strainers. Size-selected organoids were centrifuged again for 5 min at 4 °C, and their density was measured and adjusted for seeding at 250 organoids per well distributed in four drops (5 µL/drop). The complete MW24 assay images are shown in Appendix A. Upon Matrigel solidification, medium was added, and organoids were incubated at 37 °C for 24 h before drug treatment. Seven drug concentrations, plus DMSO (vehicle) as a control, were added and thoroughly mixed with the culture medium. Three wells were treated with each drug dose (triplicates). After incubation for 4 days, ATP was measured using the Cell Titer-Glo 3D Cell Viability Assay kit (Promega) with slight modifications. Briefly, culture medium was removed and cells washed with PBS before addition of fresh 100 µL PBS/well. Matrigel drops were detached with a flat spatula, and 100 µL of Cell Titer-Glo 3D Cell Viability assay reagent were added per well. After 5 min incubation on an orbital shaker followed by 25 min rest (both in the dark), 90 µL aliquots from each well were loaded (in duplicate) in a 96-well luminometer plate and measured three times in a Glomax 96 luminometer (Promega). For each patient, cytotoxicity assays were performed at least three times at different passages using triplicates.

### 4.11. Wash-Out Assays

The procedure was identical to the cytotoxicity assays except that 4 hours after drug treatment, the organoid drops were washed with PBS and tumor organoid growth medium was added. After incubation for 4 days, cell viability was measured as described for cytotoxicity assays.

## Figures and Tables

**Figure 1 marinedrugs-17-00648-f001:**
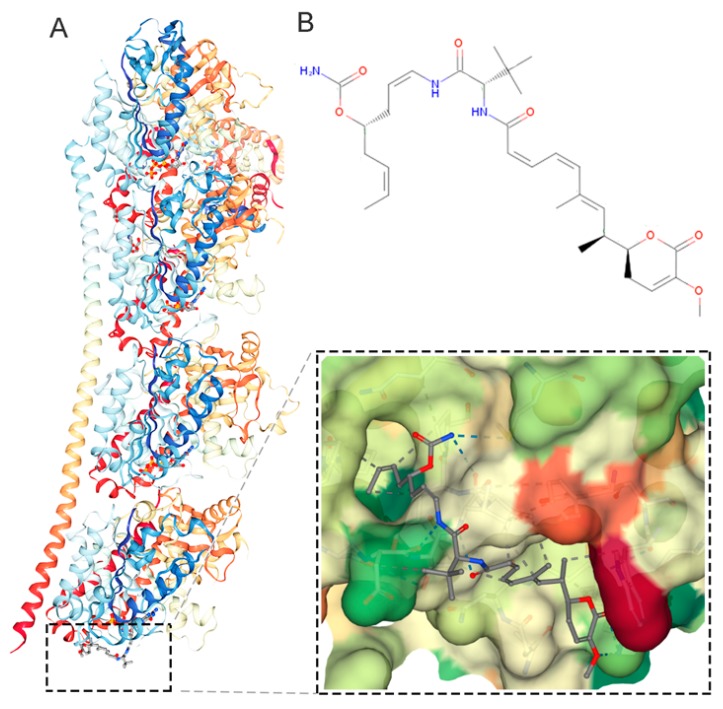
Plocabulin action and structure. **A**, Model of the interaction between plocabulin and tubulin (Protein Data Bank ID: 4TV9) [6] using Jmol [8] and NGL viewers [9] **B**, Chemical structure of plocabulin (Chemspider ID: 30771013).

**Figure 2 marinedrugs-17-00648-f002:**
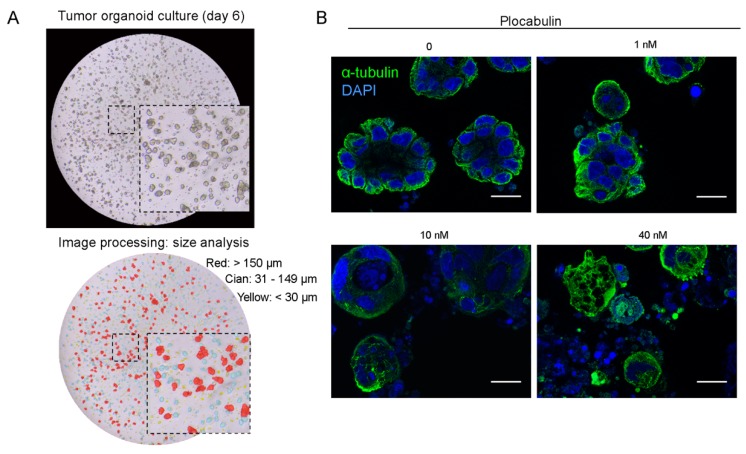
Tumor organoid culture and plocabulin activity pilot assay. **A**, Phase-contrast micrographs of patient-derived organoids of heterogeneous sizes. Upper panel, images 6 days after seeding. Lower panel, images after MIPAR processing (see Materials and Methods) showing small (<30 μm, yellow), medium (31–149 μm, cian), and large (>150 μm, red) organoids derived from patient #3 biopsy. **B**, Immunofluorescence analysis of α-tubulin (green) and DAPI (4′,6-diamidino-2-phenylindole, blue) in colon tumor organoids treated with the indicated concentrations of plocabulin for 4 days. Scale bar, 16 μm.

**Figure 3 marinedrugs-17-00648-f003:**
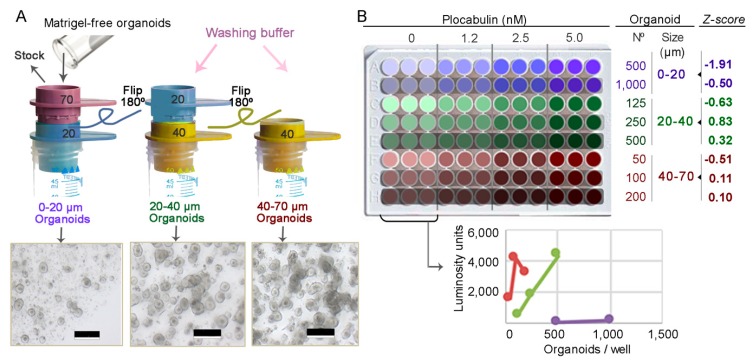
Optimization of drug activity assays. **A**, Procedure for establishing cultures of size-selected organoids. Three cell strainers were used to obtain organoids of 0–20 μm, 20–40 μm, or 40–70 μm diameter. Micrographs show the organoid cultures 4 days later. Scale bars, 200 μm. **B**, Representative optimization assay for patient #4, passage 24. Size-selected organoids were seeded in triplicate in eight combinations of different organoid sizes and seeding densities and were treated with three plocabulin concentrations or vehicle (0). The plot shows cell viability of controls (luminosity units) in relation to organoid size and number of organoids seeded/well. Medium-sized organoids (green) showed a linear relationship between luminosity response and seeded organoid number with a good Z-score, whereas small organoids (purple) generated insufficient signal, and larger organoids (red) showed a non-linear relationship (saturation/incomplete lysis) and lower Z-scores. Z-score was used as a quality control parameter to ensure adequate drug-response detection.

**Figure 4 marinedrugs-17-00648-f004:**
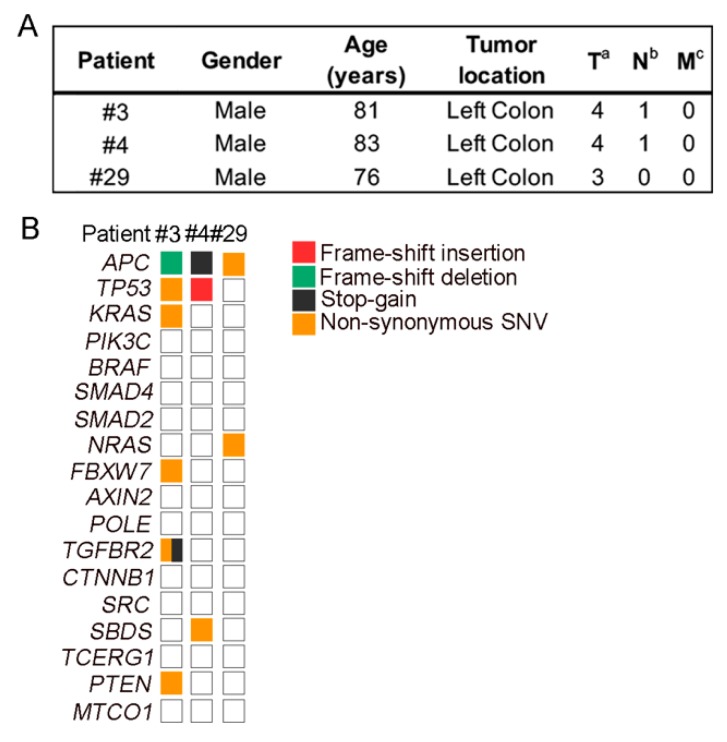
Patient and organoid characteristics. **A,** Clinicopathological data from CRC (colorectal cancer) patients. **B**, Overview of the mutations found in tumor organoid cultures.

**Figure 5 marinedrugs-17-00648-f005:**
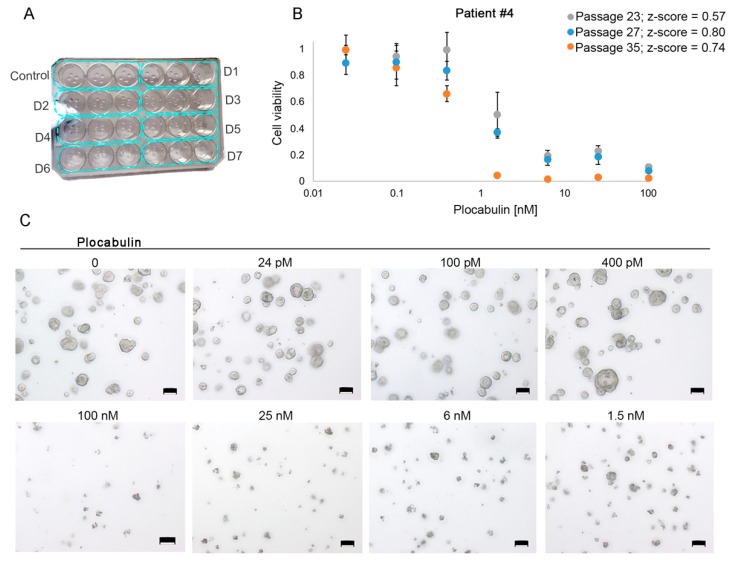
Cytotoxic assays testing plocabulin and quality control by Z-score analysis. **A**, Design of a cytotoxic assay. Four Matrigel drops containing organoids are seeded per well in MW24 plates, and triplicates are run for each drug dose (blue rectangles). D, dose. **B**, Plocabulin dose-responses after 4-day treatment for patient #4 in three independent assays (passages #23, #27, and #35). For all assays, Z-score >0.5. Data are mean ± SD of the triplicate runs for each independent assay (different passages). **C**, Representative phase-contrast images of one cytotoxic assay for patient #4 (dose-response curve in B). Scale bars, 100 μm.

**Figure 6 marinedrugs-17-00648-f006:**
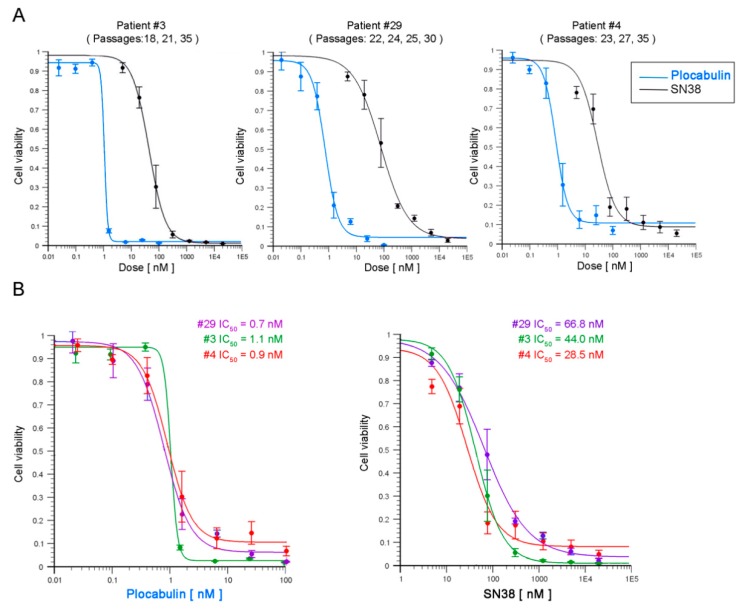
Cytotoxic activity of plocabulin in colon tumor organoids. **A**, Dose-response curves for plocabulin (blue lines) and SN38 (black lines, (7-Ethyl-10-hydroxy-camptothecin)) in the three CRC–patient-derived organoid cultures. Data are shown as mean ± SEM of independent assays with organoids at different passages. For all assays, Z-score >0.5. **B**, Overlaid dose-response curves for plocabulin and SN38 (patients #3, #4 and #29) and calculated IC_50_ values.

**Figure 7 marinedrugs-17-00648-f007:**
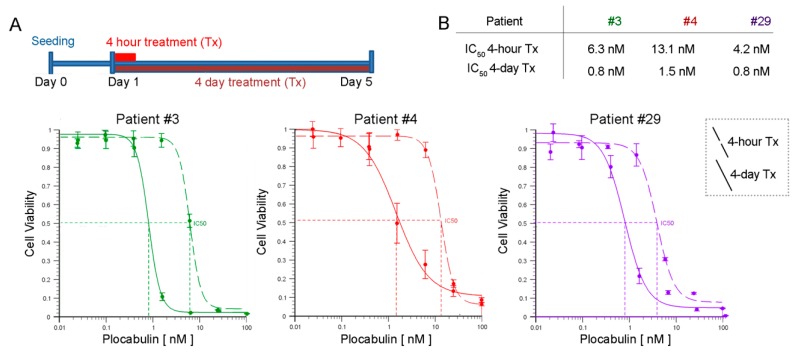
Plocabulin activity in wash-out assays. **A,** Plocabulin cytotoxic activity was compared between continuous treatment (4 days) and pulse treatment (4 hours). Graphs show dose response curves for continuous treatment (solid lines) and pulse treatment (dashed lines) in each patient-derived organoid culture. All experiments used 20–40 μm diameter organoids seeded at 250 organoids/well. **B**, IC_50_ values and individual dose-response curves were calculated using the AAT Bioquest IC_50_ calculator [24]. Data are mean ± SD of triplicate determinations.

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
