# Peer review of "Plocabulin Displays Strong Cytotoxic Activity in a Personalized Colon Cancer Patient-Derived 3D Organoid Assay"

_marinedrugs, 2019, doi:10.3390/md17110648_

Round 1
Reviewer 1 Report
Recommendation: Major revision
Comments to Author:
Title: Plocabulin displays strong cytotoxic activity in a personalized colon cancer patient-derived 3D organoid assay
Authors: Alba Costales-Carrera, Asunción Fernández-Barral, Pilar Bustamante-Madrid, Laura Guerra, Ramón Cantero, Antonio Barbáchano and Alberto Muñoz
Overview and general recommendation:
The manuscript by Costales-Carrera, Muñoz et al. entitled “Plocabulin displays strong cytotoxic activity in a personalized colon cancer patient-derived 3D organoid assay” presents the application of colon cancer organoids to test the cytotoxic activity of a novel drug (PM060184), actually in phase 2 of clinical trials. The work is novel and well structured. In the results section (text) the authors described politely and comprehensively the 3D organoid culture and Plocabulin cytotoxic activity, however for the reviewer needs an in-depth revision of figures. If the manuscript will be revised by taking the following comments into account, reviewer would be willing to recommend this paper for publication in Marine drugs. The comments were listed as below.
Although the introduction paragraph is well structured, it should be revised emphasizing the previous findings published by the group. As extended reported in literature (Cell Stem Cell 2016, 18, 827–838), an organoid in order to be such should be characterized by gene expression and xenografts to demonstrate the recapitulation of the original clinical phenotype. In the present manuscript the authors presented a thin genetic analysis while is missing the xenograft to prove the tumorigenicity. The reviewer suggests emphasizing the previous work already at the end of the introduction, maybe inserting the ref 19 at page 2 line 63. The Figure 1 in the result section is unclear and might confuse the readers. The text is too small and difficult to see, especially the scale bars and plots of figure 2D. In particular, the authors assembled in the same panel the characterization of the organoids and the drug assay. The reviewer suggests reorganizing Figure 2 into two new figures. The first one (novel figure 2) to show the organoids characterization and method of selection, with previous figure 2 (A) and (C). The second (novel figure 3) to show the drug assay assembling figure 2 (B) and (D). The panel with genetic alteration (Figure 3) could be inserted between the two new figures. Figure 4 and relative legend is also unclear. What means passage? The authors should clarify this aspect in the manuscript. In the text (pag 4 line 116) the authors stated that “organoid cultures at three different passages were incubated with increasing doses of plocabulin for 4 days” while in the legend of figure 4 the days became 6. What is the duration of the drug treatment? If is 4 days, why this duration has been chosen? The authors should explain in a clear way this part. Minor problems of figure 4: The scale bars are too small and the reviewer suggest to remove arrows. Usually, at a given concentration of a given anticancer drug on a given cell line/spheroid/organoid, the viability of these constructs will decrease with increasing incubation times of the cells with the drug. In general, is applied an incubation time from 24 to 72h. In the present work, the authors tested only one incubation time that was compared to the wash out assay. The reviewer suggests to test a intermediate incubation time (i.e. 48h) to compare the obtained data and increase the impact of the work. To improve this manuscript, the discussion section should be revised especially extending and commenting the findings obtained in the cytotoxicity assays, the dose and time response of the drug. Minor drawbacks: There are some typos and grammatical errors. Authors should check the manuscript carefully.Author Response
Please see the attachment.

Reviewer 2 Report
The potency of plocabulin was evaluated in 3D organoid cultures that were set up from three colon cancer patient-derived cells. The activity of plocabulin was compared with that of SN38 in the organoid cultures in a 4 day continuous- or pulsed- exposure. The data showed reproducibility across three-four passages of organoid cultures. It would have been interesting to also compare the dose-responsiveness and reproducibility of plocabulin in routine 2D cultures of colorectal cell lines with similar genetic background. For prioritizing hits from primary drug screens, the study could have been expanded to evaluating responsiveness to drugs of varying potencies in the 3D cultures.
Reviewer 3 Report
Reviewer comments to manuscript: marinedrugs-636493
Background: The authors developed a new platform using a 3D-cancer model (based in the formation of small organoids) using patient-derived cells (from biopsies of colorectal cancer [CRC] patients) to test the effect of a novel antitumor agent (Plocabulin) with specific activity of disrupting cellular microtubule organization. Interestingly, this novel molecule was originally purified from the marine sponge Lithoplocamia lithistoides and is now available in its synthetic form, which was used by the authors. In addition, it was demonstrated previously that Plocabulin is a potent antiproliferative and antiangiogenic drug in in vitro assays using 2D standard cultures of carcinoma cell lines as well as in xenograft models performed in mice. Specifically, in this work the authors demonstrated that Plocabulin is an order of magnitude more potent than the irinotecan derivative SN38 (which is the current drug used to treat CRC affected patients) when using colon cancer organoids. Their results, in fact, support undergoing clinical trials (phase II). This clearly indicates the value of the research work performed by the authors in providing a proof of concept for their organoid model to pre-test anticancer drugs in a personalized fashion before patent treatment.
Recommendation: In my opinion the manuscript is the demonstration of a well-performed work, without methodological or rational errors, wrong data interpretation or misleading conclusions. In addition, is well written and clear. The authors are well focused and clearly demonstrated that their organoid model can be used for future pre-test of patient biopsies. Nevertheless, I have some suggestions and questions (please, see comments below) since in some parts the manuscript could be improved. Please, follow my recommendations since I believe if corrected or clarified you would improve the present version of your work.
Comments:
This is a question based in my limited knowledge of the colorectal cancer area and specifically in the potential availability of cell types, cell lines or cell derivates that can be used as normal (or less aggressive) controls. It is possible to obtain organoids, or some type of normal-(non-carcinogenic) 3D-culture model in order to test how active Plocabulin on normal cells? This, if not possible, should be discussed somewhere, in order to indicate to the reader that this anti-tumor agent is specific for cancer (in principle malignant cells) and no so toxic (or no toxic) to normal cells. Moreover, in terms of the patient´ biopsies: do these samples carry also normal cells that can be used as controls?
It is not clear why you need to grow the organoids in absence of Wnt3A, R-spondin and nicotinamide. Somehow seems that these factors are secreted by normal surrounded tissues. In any case, should be better indicated to be sure the reader can understand better the methodological procedure and why.
2B. Why Beta-tubulin is mainly detected at the periphery of the organoid? At least in control and 1 micromolar concentration of Plocabulin. This is because is strictly expressed there or because the detection of the internal Beta-tubulin is difficult by the fluorescent microscopy method used?
2A and B. It is clear that, by reading M&M that you develop organoids by resuspending dissociated cells from the biopsies into Matrigel. Could you form organoids by simple growing cells on non-adherent dishes or by hanging drop technique? The second question is: The size distribution of organoids (small, medium and large) are due to 1, heterogeneity in the initial number of cells seeded which then, as consequence, will bring different size organoids (meaning that you might have single cells, double cells, or small initial clusters, etc.) or 2, your initial seeding are single cells but with three different cell types having different cell division kinetics which ends in the formation of three size of organoids. If the latest is true, by selecting medium size organoids (for obvious reasons of linear correlations) it is also possible that you are selecting a group of cells with medium-proliferation capacity and not considering very low- (may be more normal) or very high-proliferative (may be more aggressive) in your assays? In any case, if 1 or 2, or a combination of both is happening, it should be clarified and discussed. One solution in the future could be to use medium-size organoids (because the measuring is in the linear range) but isolate each type at different times. For instance, the small ones can be maintained for longer until their reach medium size, and so on. In this way, regardless of the cells division kinetics you always use medium-size organoids.
In relation with Fig. 3. You wrote in the text (line 109) that KRAS was (2/3) but if I understood properly the Fig. 3 it should be (1/3). Please, clarify.
By looking at Fig. 4 A, seems that in this case the organoids used are a mix of small, medium and large ones. If I understood correctly in previous results it was defined to use only medium-size organoids. Did you select medium-size or you used the total initial mix? Please, clarify.
In Fig.5, cell viability is plotted in function of drug dose of Plocabulin and SN38 for the three patient-derived samples. In brackets number of passages is indicated for each of the three graphs. My question is: Which passage correspond to the graph below? Or is a combination of all the passages? It should be clarified unless I am not understanding it, which could be the case.
If you know, there is any way to test in vitro the potential of Plocabulin, or any other drug in the future, with normal cell models? You can discuss this/these possibilities. May be in animal tests is clear that normal tissues did not suffer the effects observed in xenografts systems in mice. Please, clarify.
Round 2
Reviewer 1 Report
The authors have satisfactorily addressed all comments. The paper is publishable on Marine Drugs journal.